

# Preliminary placement and new records of an overlooked Amazonian tree, *Christiana mennegae* (Malvaceae)

Rafael G. Barbosa-Silva[1,2], Thales Silva Coutinho[3], Santelmo Vasconcelos[1], Delmo Fonseca da Silva[4], Guilherme Oliveira[1] and Daniela C. Zappi[1,2,5]

[1] Instituto Tecnológico Vale de Desenvolvimento Sustentável, Belém, Pará, Brazil
[2] Coord. Botânica, Museu Paraense Emílio Goeldi, Belém, Pará, Brazil
[3] Programa de Pós-graduação em Biologia Vegetal, Departamento de Botânica, Universidade Federal de Pernambuco, Recife, Pernambuco, Brazil
[4] Parque Zoobotânico, Departamento de Ferrosos Norte, Gerência de Meio Ambiente–Minas de Carajás, Parauapebas, Pará, Brazil
[5] Programa de Pós-graduação em Botânica, Instituto de Ciências Biológicas, Universidade de Brasília, Brasília, DF, Brazil

Corresponding author
Rafael G. Barbosa-Silva,
rafa.g29@gmail.com

## ABSTRACT

*Christiana mennegae* is a phylogenetically enigmatic taxon and represents a case in point of a species whose presence escaped the radar of the Amazon lists and the Brazilian Flora project. Here we expand its distribution by adding new records from Peru and overlooked ones from Brazil. To investigate its phylogenetic placement in the Brownlowioideae, part of the rbcL gene of the plastid and the intergenic ITS2 region were sequenced. Macro- and micro-morphological investigation of features of *C. mennegae* using SEM of foliar, flower, fruit and seed structures are presented. A lectotype for the name is designated here. The morphology of trichomes revealed five types of trichomes ranging from glandular to branched and unbranched and we also report stomata on the seed surface for the first time in Brownlowioideae. *Christiana mennegae* and *C. africana* were recovered as sister species in the phylogenetic analysis, albeit with low to moderate support, and more species of this and closely related genera must be sampled and analyzed in order to obtain a clearer picture of the group's affinities and relationships. We provide an update of its conservation status from Vulnerable to Least Concern. We also highlight the need for investment in the digitization of biological collections, botanical capacity building at the local level and the importance of the availability of online literature to speed the study of Amazonian plant diversity.

## INTRODUCTION

The exploration of the Amazon rainforest has sparked scientific curiosity since the travels of naturalists such as Alexandre Rodrigues Ferreira, La Condamine, Spix and Martius, Schomburgk, Spruce, Wallace and Bates (*La Condamine & Godin des Odonais, 1813*; *Spruce, 1908*; *Ferreira et al., 2008*; *Spix & Martius, 1824*; *Hemming, 2015*; *Schomburgk, 1922*). The stunning variety of life forms, coupled with the difficulty of accessing vast areas of the biome has led to repeated efforts to quantify its diversity

(*Black, Dobzhansky & Pavan, 1950*; *Pires, Dobzhansky & Black, 1953*; *Gentry, 1988*; *Hubbell et al., 2008*; *Milliken et al., 2010*; *ter Steege et al., 2016*; *Cardoso et al., 2017*). Recent studies indicate that the lowland Amazon region is home to 14,003 seed plant species (*Cardoso et al., 2017*). Despite the global importance of the Amazon Forest, the information deficit is still high, and knowledge is fragmentary, with vast areas still poorly known (*BFG, 2018*; *Hopkins, 2019*), and some of them already destroyed (*Piontekowski et al., 2019*). The number of species based on herbarium data and/or large online datasets has changed little and spatial collecting gaps persist (*Forzza et al., 2012*; *BFG, 2015*; *Hopkins, 2019*).

Attempts to accelerate the work to describe the Amazon's plant diversity include the development of massive online data repositories (*Canhos et al., 2014*; *Morim & Lughadha, 2015*; *BFG, 2018*). Both herbarium collections and literature sources have been included in this endeavor (*Ebach, Valdecasas & Wheeler, 2011*; *Funk, 2018*) and, in the age of big data, a plurality of sites and online libraries help to minimize this taxonomic impediment (*Schonfeld, 2003*; *Gwinn & Rinaldo, 2009*). However, botanical studies are still scarce and do not cover the taxonomic complexity of a megadiverse biome such as the Amazon (*Hopkins, 2019*). Access to existing studies is essential for the advancement of this knowledge.

*Christiana* DC. (Malvaceae) is included if the subfamily Brownlowioideae on phylogenetic evidence and due to morphological features such as sepals fused to form a campanulate calyx, stamens numerous, free, anthers basally dilated with apically contiguous thecae and many-flowered bicolor units (*Bayer et al., 1999*; *Bayer & Kubitzki, 2003*). The genus comprises five species worldwide, four of them tropical, and was described by *De Candolle (1824)*. The type of the genus, *Christiana africana* DC., is a tropical tree with similar distribution to the kapok tree (*Ceiba pentandra* (L.) Gaertn.). A second species, *C. macrodon Toledo (1952)*, was described from the Atlantic Rainforest of the state of São Paulo. *Kubitzki (1995)* widened the circumscription of the genus to include two other Neotropical species (*C. mennegae* (Jans-Jac. & Westra) Kubitzki and *C. eburnea* (Sprague) Kubitzki, as well as the Polynesian *C. vescoana* (Baill) Kubitzki.

Despite the fact that *Kubitzki (1995)* mentioned the occurrence of *Christiana mennegae* in Brazil, this was overlooked by successive editions of the Catalog of Brazilian Plants and Fungi (*Forzza et al., 2010*), List of Species of the Brazilian Flora (Lista de Espécies da Flora do Brasil, 2008–2015), BFG 2015 (*Brazil Flora Group [BFG], 2015*) and the Brazilian Flora 2020 (*Flora do Brasil, 2020*). In addition, *Secco (2000)* recorded *Asterophorum mennegae* Jans.-Jac. & Westra (= *C. mennegae*) for the state of Pará, although his publication was also overlooked. *Christiana mennegae* is not listed in the Amazon tree list (*ter Steege et al., 2019*), nor was it recorded in the list of Amazonian seed plant species (*Cardoso et al., 2017*). Meanwhile, this species has remained as Vulnerable in the Red List since 1998 (*World Conservation Monitoring Centre, 2017*).

Morphological studies with other genera of Brownlowioideae have explored characteristics such as seed coat and the type of trichome on the leaf surface for the delimitation of species (*Chung, Tan & Soepadmo, 2012*), and such characters were still unknown for *Christiana*. In addition, phylogenetic sampling of the Brownlowioideae is still limited to only one species per genus, with the genus *Christiana* represented only by *Christiana africana* DC. (*Bayer et al., 1999*; *Hernández-Gutiérrez & Magallón, 2019*). Thus,

*Christiana mennegae* is another rare, phylogenetically enigmatic and poorly understood tree species from the Amazon (*Cardoso et al., 2015*). During fieldwork linked to a project to study the natural capital of the Amazon forest in the region of Carajás, our attention was drawn to the existence of an unidentified malvaceous tree by DS, one of the authors of this paper, at the Parque Zoobotânico in the Floresta Nacional de Carajás (FLONA Carajás) that turned out to be *C. mennegae*. The specimen collected in the area had morphological characteristics that matched the genus *Christiana*, such as flowers with a fused calyx but with free lobes, numerous stamens and fruits with glabrous and glossy endocarp, carrying a single, variegated seed per locule.

With the purpose of clarifying and complementing the knowledge regarding *Christiana mennegae*, we provide additional data on its morphology and phylogenetic affinities to establish with certainty its presence in Brazil and Peru.

## MATERIAL AND METHODS

### Taxonomic treatment

The morphological description was mostly based on herbarium material supplemented by measurements from living plants collected in recent expeditions to the Floresta Nacional de Carajás (FLONA Carajás/63324-1), Pará state, Brazil. Several branches of the specimens were examined in the field to be sure that the species is dioecious. Herbarium material was rehydrated to dissect and illustrate reproductive structures. We analysed specimens from the HCJS and MG and searched the web for records available in *REFLORA (2020)*, *Tropicos (2020)* and Virtual Herbarium of Flora and Fungi of Brazil (*SpeciesLink network, 2020*). We found specimens deposited in F, IAN, INPA, K, MG, MO, P, RB, U and VEN herbaria (acronyms according to *Thiers, 2020*), continuously updated). Morphological terminology follows *Harris & Harris (2001)* and *Radford et al. (1974)*.

### Distribution map and conservation status

The distribution map was prepared using QGIS (*QGIS Development Team, 2016*) with coordinates obtained based on information available from herbarium labels. All studied specimens were plotted on the map. The IUCN conservation status (*IUCN, 2012*) was ascertained using the GeoCat Tool (*Bachman et al., 2011*).

### SEM

Petioles, abaxial surface of mature leaf-blades, external surface of the calyx, epicarp and seed were dried and extracted from each voucher specimens deposited at MG (Zappi et al. 4562 and Barbosa-Silva et al. 1424). These structures were mounted on stubs using press-on, double-sided tape tabs, examined with a Scanning Electron Microscope (SEM) and imaged in Zeiss Sigma VP microscope under full vacuum at the Laboratório de Microanálises of the Universidade Federal do Pará (IG - UFPA). It was not necessary to critical-point dry them. The trichomes were described according to *Theobald, Krahulik & Rollins (1979)*. The images were falsely colored using Photoshop CS4 (Adobe).

## Sampling, DNA extraction, PCR amplification, sequencing and phylogenetic analysis

For this study, the subfamilies Bombacoideae, Brownlowioideae, Byttnerioideae, Dombeyoideae, Grewioideae, Helicterioideae, Malvoideae and Sterculioideae are represented by sequences published by *Hernández-Gutiérrez & Magallón (2019)*. *Christiana mennegae* was newly sequenced for this study and included in the aligned matrix (Zappi et al. 4562 - MG). The outgroup included species of Bixaceae, Cistaceae, Neuradaceae and Thymelaeaceae.

We reconstructed a Malvaceae *s.l.* phylogenetic tree in order to infer the placement of *Christiana mennegae* using five molecular regions, four from the plastid genome (*rbc*L, *atp*B, *trn*K-*mat*K, *ndh*F), and the nuclear ITS2 from the 35S rRNA. For *C. mennegae*, sequences of ITS2 and *rbc*L were newly obtained and deposited in the GenBank database under the accession numbers MT741784 and MT742086, respectively. For a detailed description of automated DNA extraction protocol see electronic supplementary material.

The alignment of *Christiana mennegae* DNA sequences was performed manually in the aligned matrix by *Hernández-Gutiérrez & Magallón (2019)*. Bayesian Inference (BI) and Maximum Likelihood (ML) methods were used for the phylogenetic reconstruction. Partitioned BI analyses were performed using MrBayes version 3.2.3 (*Ronquist et al., 2012*), with DNA substitution models selected by *Hernández-Gutiérrez & Magallón (2019)* (GTR + I + G). Further details on phylogenetic analysis can be also found in electronic supplementary material.

## RESULTS

### Taxonomic treatment

***Christiana mennegae*** (Jans.-Jac. & Westra) Kubitzki, Bot. Jahrb. Syst. 116 (4): 541. 1995. ≡ *Asterophorum mennegae* Jans.-Jac. & Westra, Proc. Kon. Ned. Akad. Wetensch, B 86(3): 377. 1983.

**Type.** Suriname, [Sipaliwini], "Morro Grande" camp-forest island, 6 km W of "Morro Grande" dome, Sipaliwini savanna area on the Brazilian frontier, 04 nov 1968 (lf, fl, fr), *F.H.F. Oldenburger, R. Norde & J.P. Schulz ON415* (second-step lectotype, designated here: U [U0006902; digital image!], isolectotype: K [K000381142; digital image!], MO [not seen], NY [NY00415374; digital image!], P [P02143012; digital image!], U [U0006903; digital image!], VEN [VEN409940; digital image!]). (Figs. 1, 2, 3, 4 and 5).

**Description.** *Trees* 5–17 m tall, functionally dioecious; external bark in male plants shedding as thin scales, female plants with conspicuous and raised lenticels. *B ranches* cylindrical, lenticellate, with sparse stellate trichomes. *Stipules* 2.8–3.8 mm long, linear, apex acute, with stellate trichomes. *Leaves* alternate, spirally arranged; petioles 1.8–5.6 cm long, with scattered minute stellate, sessile trichomes; leaf-blades chartaceous, 12.3–22(–29) × 5–8(–9) cm, narrowly elliptic, lanceolate or oblanceolate, apex long acuminate, caudate or cuspidate, base rounded to obtuse, rarely slightly subcordate, margin entire or rarely slightly erose, adaxial surface shiny when dried, glabrescent except for scattered stellate, sessile trichomes on the veins, abaxial surface glabrescent or with scattered glandular,

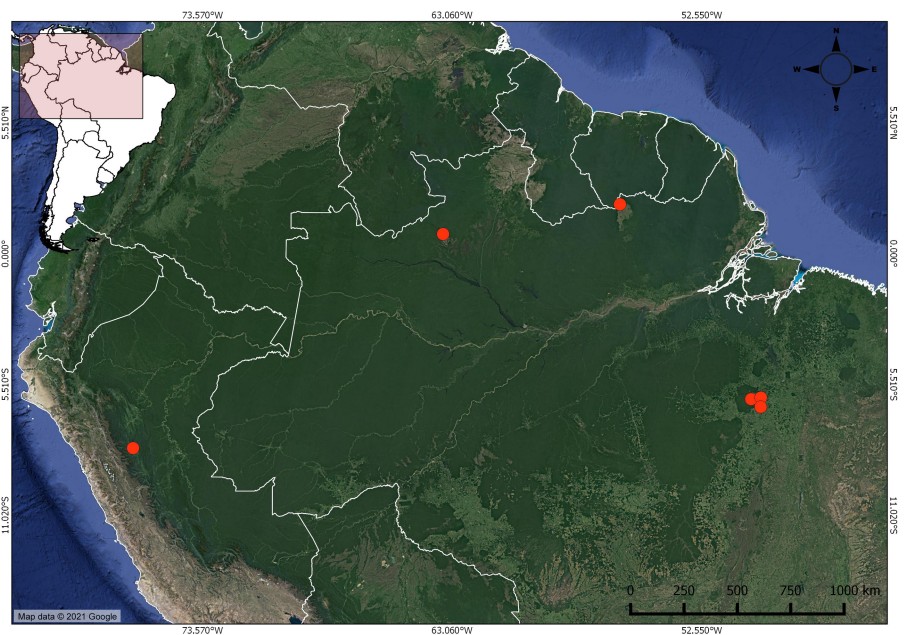

**Figure 1** **Distribution of *Christiana mennegae* based on examined material.** In the far west, the new record of *C. mennegae* in Peru.

trichomes stellate on the veins; venation eucamptodromous, 5–6 pairs of secondary veins, veins abruptly spaced out distally. *Inflorescence* axillary, congested or laxa, 3.6–4(−5.5) cm long, 3–8-flowered cymes; stellate trichomes, bracts 3–1.2 mm long, triangular, with lateral linear projections, stellate trichomes on both surfaces, bracteoles 2–2.7 × 0.3–4 mm, entire, linear, stellate trichomes. Flowers functionally unisexual, pedicellate; calyx gamosepalous, fused by 2 mm long, with free lobes, externally green, with stellate trichomes, internally glabrous, subglobose to campanulate; corolla dialypetalous, white, petals obovate to oblanceolate or elliptic, glabrous on both surfaces. *Staminate flowers:* pedicels 2.5–4 mm long; flowers ca. 8 mm long; calyx (3–)4–merous, c. 2.2 × 3 mm; corolla 5–7-merous, petals 5.5–7 × 1.8–3.5 mm; stamens 63–67, the inner ones longer than the outer ones, glabrous, filaments 2.5–3 mm long, connate by c. 1 mm long, anthers 0.5–0.8 mm long. *Pistillate flowers*: pedicels c. 4 mm long; flowers c. 6(−7) mm long; calyx (3–)4–5-merous, 1.5–2.8 × 1.5–2(–4) mm; corolla 5–7-merous, petals 5–7.5 × 2–2.8 mm; staminodes 2–2.7 mm long, anthers c. 0.3 mm long; gynoecium c. 4.2 mm long, ovary c. 2 × 2 mm, trichomes stellate, style c. 1.8 mm long, trichomes stellate, stigma 5, c. 0.6 mm long, ovules c. 0.5 mm long. *Fruit capsule*, woody, syncarpous, 1.2–1.8 cm long, c. 3.8 cm diam., depressed-globose, externally with stellate trichomes, composed of 5 loculicidal, one-seeded per locule; seed obovate, 8–9 × 6–7 mm, glabrous, medium brown with darker irregular marks.

**Nomenclatural notes:** *Christiana mennegae* was described from a single collection deposited at U, *Oldenburger ON415*. However, there are two separate sheets from this collection in the same herbarium. *Jansen-Jacobs & Westra (1983)* did not specify which sheet should be the type, and although they published photos, they use images of the two

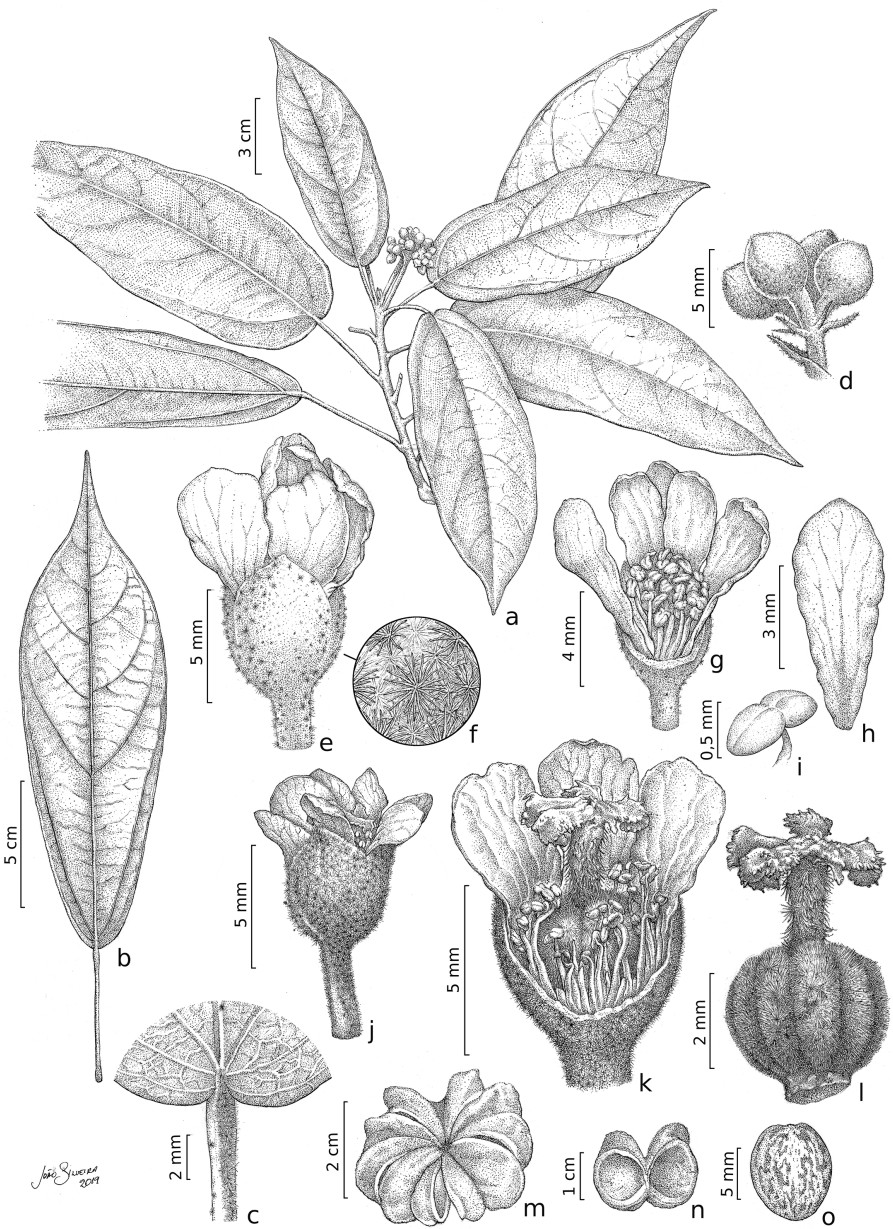

**Figure 2** *Christiana mennegae.* (A) Fertile branch. (B) Leaf. (C) Detail of the petiole showing stellate trichomes. (D) Detail of an inflorescence. (E) Staminate flower. (F) Detail of the stellate multi-angulate trichomes on the calyx. (G) Staminate flower with a petal removed showing the arrangement of the stamens. (H) Petal (adaxial view). (I) Anthers. (J) Pistillate flower. (K) Detail of a dissected pistillate flower with staminodes. (L) Gynoecium. (M) Fruit. (N). Side view of capsule valves. (O) Seed. A–I from *Zappi et al. 4562*; J–O from *Barbosa-Silva et al. 1424*. Drawn by João B. Silveira.

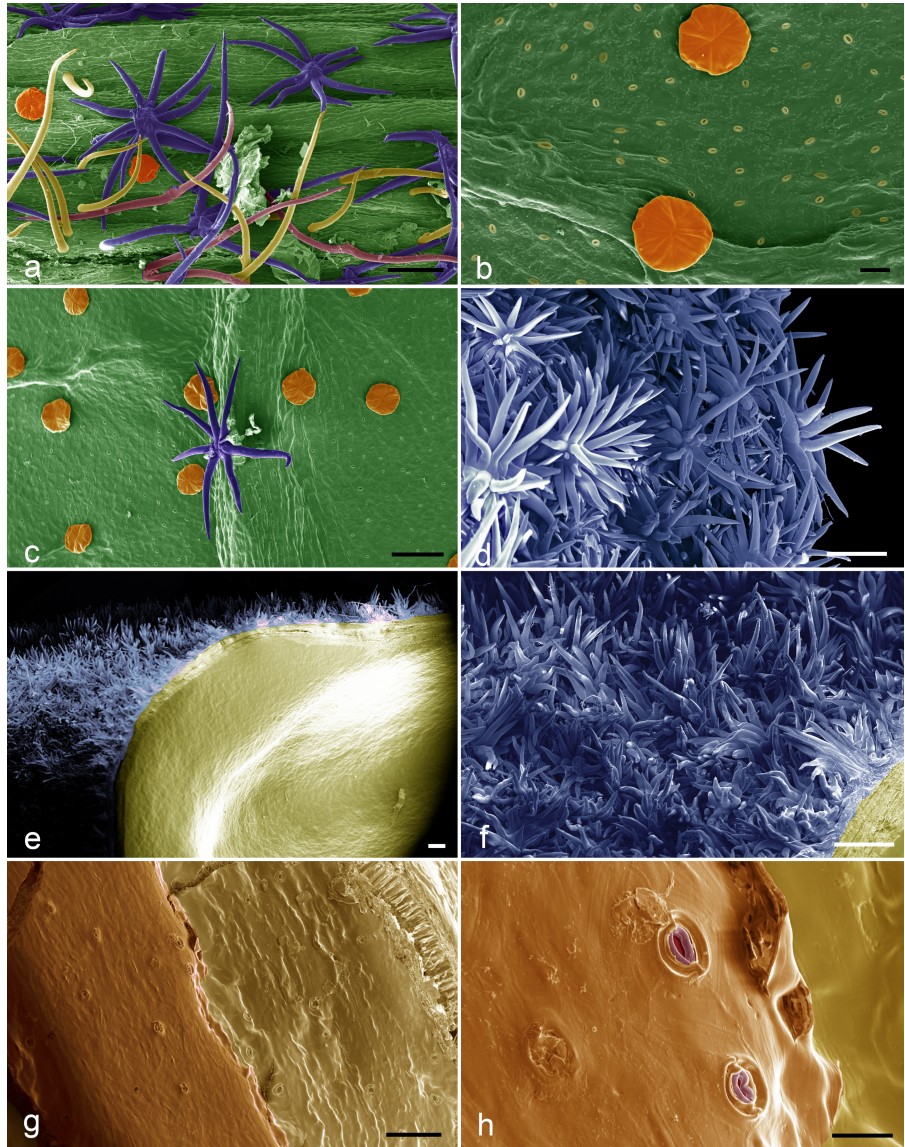

**Figure 3** **SEM images of the *Christiana mennegae*.** (A) Simple trichomes (yellow), stellate-rotate trichomes (dark blue), two-armed trichomes (pink), glandular trichomes (orange) on the petioles. (B) Glandular trichomes (orange) and stoma on abaxial surface of the blade leaf. (C) Stellate rotate trichomes (dark blue) on the veins of the abaxial surface of the blade leaf and glandular trichomes (orange) on the rest of the surface. (D) Stellate-multiangulate trichomes on external surface of the calyx. (E) Fruit showing stellate-multiangulate trichomes (light blue) on epicarp and glabrous and without stomata on endocarp (yellow). (F) Detail of concentrated stellate multiangulate trichomes (light blue) on epicarp. (G) Seed surface showing many stomata. (H) Details of stomata in seeds surface. Scale bars: A-100 μm; B- 20 μm; C- 100 μm; D- 100 μm; E- 200; F- 100 μm; G-100 μm; H-20. μm. (A–I from *Zappi et al. 4562*; J–O from *Barbosa-Silva et al. 1424*. Images by Ana Correa.

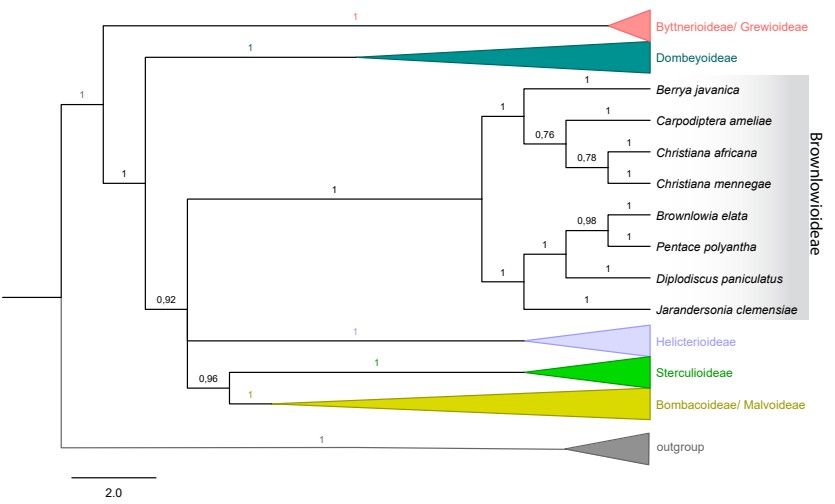

**Figure 4** **Majority rule consensus tree of Malvales.** Emphasis on subfamily Brownlowioideae resulting from a Bayesian analysis of the combined dataset (*atp*B, *trn*K-*mat*K, *ndh*F, *rbc*L and ITS).

materials, considering both as the holotype. Therefore, it is necessary to do the second step of lectotypification and we selected specimen U0006902 because it is more complete than the other sheets.

**Distribution and habitat:** *Christiana mennegae* was first known from Suriname, Brazil (*Jansen-Jacobs & Westra, 1983*; *Kubitzki, 1995*; *Secco, 2000*) and here we record it for the first time in Peru. This new record is a significant extension of its range, nearly 1,700 km from the nearest recorded occurrence of the species (Fig. 1). This species can be found growing in island of evergreen seasonal forest in savannas in Suriname and in rainforest at up to 600 m elevation in Brazil. In Peru, *C. mennegae* grows in wet forest in the central region, from 500 to 600 m elevation. According to *Cruz & Taylor (2018)*, the central region of Peru has a complex geology and its botany is yet little explored. The specimens listed by *Secco (2000)*, deposited in IAN herbarium, unfortunately could not be found.

**Phenology:** The species apparently flowers at the onset of the wet season and has been collected with flowers and fruits in September, November and January and only fruiting in July.

**Comments:** *Christiana mennegae* shares syncarpic fruits with the Neotropical species *C. eburnea* (Sprague) Kubitzki and *C. macrodon* Toledo. A complete morphological comparison between the species of *Christiana* is found in a table in the electronic supplementary material. These species are distinguished by their narrowly elliptic, lanceolate or oblanceolate leaf-blades (*vs.* ovate or widely ovate to circular) with rounded to obtuse or rarely subcordate base (*vs.* truncate to subcordate or cordate). *Cristiana mennegae* also differs from *C. eburnea* by depressed-globose fruits (*vs.* subturbinate to turbinate) and from *C. macrodon* by leaf-blades with entire or rarely slightly erose margins (*vs.* sparsely dentate). From *C. africana*, the most widespread species of the genus (*Kubitzki, 1995*), *C. mennegae* can be easily distinguished by the leaf-blades narrowly elliptic, lanceolate

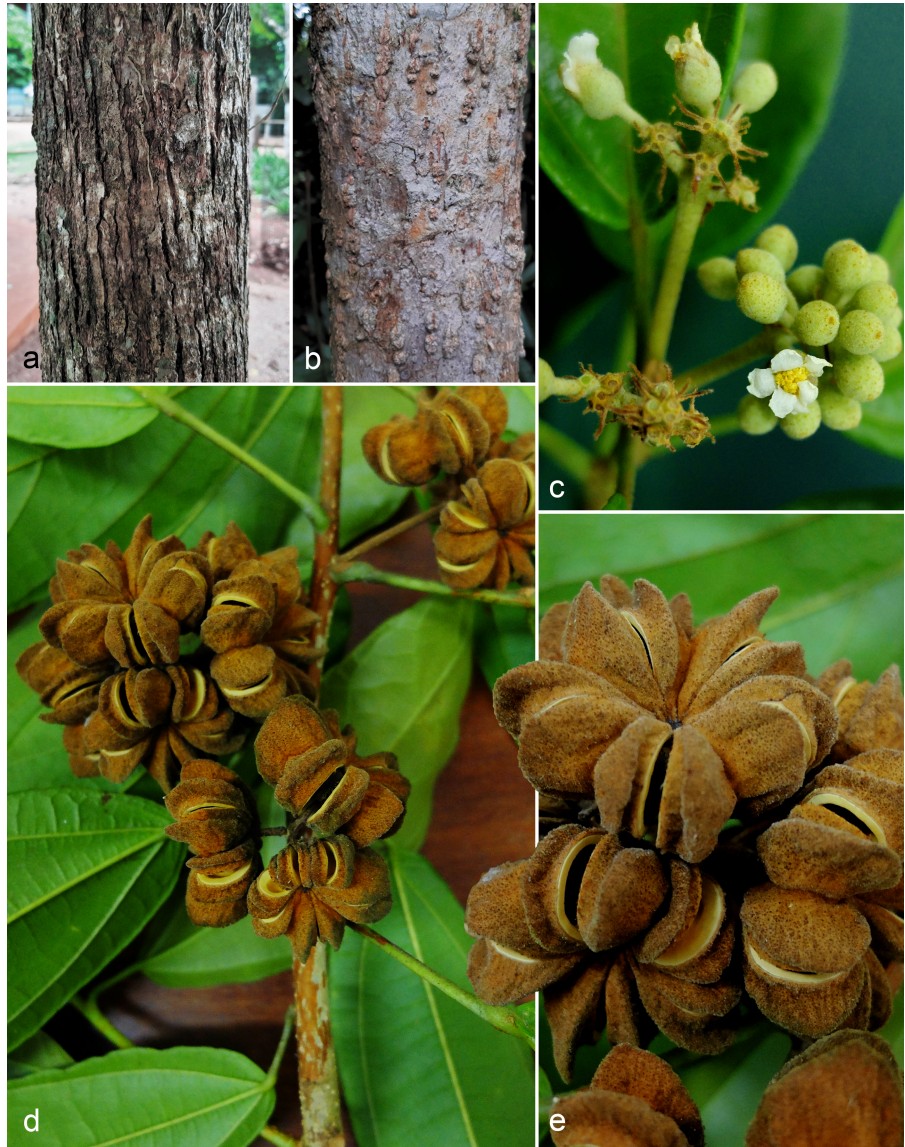

**Figure 5** *Christiana mennegae.* (A). Trunk with fissured external bark in male individual. (B) Trunk with lenticellated bark in female individual. (C) Inflorescence with buds and open flower. (D) Fruiting branch. (E) Fruit detail. A and C from *Zappi et al. 4562*; B, D and E from *Barbosa-Silva et al. 1424*.

or oblanceolate with rounded to obtuse base (*vs*. widely ovate with cordate base) and fruits syncarpic (*vs*. apocarpic) (Fig. 2). In addition, according to the results here obtained, *C. mennegae* is reported so far as the only species of the genus to present glandular trichomes on the leaf-blades, a character that distinguishes it from the other four species. The presence of up to seven petals in *C. mennegae* was not recorded until the present analysis. There were differences in appearance of the external bark between male and female individuals (Figs. 5A–5B). A total of three new physical specimens and ten digitally imaged specimens were located.

Besides the morphological characters mentioned, *Christiana mennegae* can be separated geographically from the other species. While *C. mennegae* is currently known from Suriname, northern Brazil (Amazonas and Pará states) and Peru, *C. eburnea* is known only from the type locality in Ecuador, and *C. macrodon* only from Southeastern Brazil (São Paulo state) (*Sprague, 1908*; *Toledo, 1952*; *Kubitzki, 1995*). *Christiana africana* occurs in Africa, Mexico, Central and South America (Brazil, Ecuador, Guianas and Venezuela) and Madagascar (*Kubitzki, 1995*; *Tschá, Sales & Esteves, 2002*; *Door, Jansen-Jacobs & Meijer, 2007*), however no records of sympatry between it and *C. mennegae* have been found thus far.

**Conservation status:** The last update of *Christiana mennegae* took place in 1998 and classified the species as vulnerable (*World Conservation Monitoring Centre, 2017*). The present data shows that *Christiana mennegae* occurs in two protected areas in Brazil, in the Parque Estadual da Serra do Aracá in the state of Amazonas and in the Floresta Nacional de Carajás in the state of Pará. In addition to those, it occurs in the forest islands in savanna of the Sipaliwini Savanna Nature Reserve in Suriname. Despite appearing to be locally rare, *C. mennegae* is known from four localities spread throughout the Amazon basin, with a EOO of more than 1,000,000 km$^2$ fitting within the Least Concern (LC) category of *IUCN (2012)*. Unfortunately it was not possible to estimate the size of the populations. Therefore we suggest a revision of its category of threat.

**Specimens examined:** BRAZIL. Amazonas: margem de um igarapé que nasce na Serra de Aracá, 28/07/1977, fr., *N.A. Rosa & M.R. Cordeiro 1699* (RB00059283 [digital image]). Pará: Marabá, [Mun. Parauapebas], Serra dos Carajás, 29/11/1988, fr., *N.A. Rosa & F.C. Nascimento 5084* (K001214118 [digital image]); Mun. Parauapebas, FLONA de Carajás, imediações do Parque Zoobotânico, fl./fr., 06/09/2018, *D.C. Zappi et al. 4562* (MG); Parque Zoobotânico, fl, 05/10/2019, *R.G. Barbosa-Silva et al. 1424* (MG); Nova-Canaã dos Carajás [Canaã dos Carajás], 06/01/2001, fl/fr., *L.C.B. Lobato 2624* (MG163877); PERU. San Martin: Mariscal Cáceres, Tocache Nuevo, 15/11/1972, fr., *J. Schunke Vigo 5516* (F1780039 [digital image], MO, NY).

## Micromorphology study

The five types of sessile trichomes observed in *Christiana mennegae* can be grouped under non-glandular and glandular types. There are four types of non-glandular trichomes, which can be either classed as unbranched (simple and two-armed trichomes) or fall into two different branched categories (stellate-rotate and stellate-multiangulate trichomes). A detailed description of non-glandular trichomes can be found in electronic supplementary material, micromorphology study.

The endocarp of the fruit is smooth and lacked stomata (Fig. 3E). Stomata were observed on the seed surfaces, evenly throughout the surface (Figs. 3G–3H) and were sparsely distributed in the analyzed material.

## Phylogenetic relationships

The tree topology with combined dataset (nuclear and plastid) in both analyses showed that *Christiana mennegae* emerged in a clade with *C. africana*, although only the ML analysis

resulted in a highly supported monophyletic branch (BS = 86; PP = 0.78) (Fig. 4). As there are sequences available only for these two species of the genus, we could not reach any further conclusions on the relationships within *Christiana*. In addition, our results revealed *Christiana* in a strongly supported clade including the sampled species of *Berrya* and *Cardioptera* (BS = 78; PP = 1), although with no clear relationship patterns among the genera (Fig. 4). Both analyses recovered the same supported clades (see ML tree in *electronic supplementary material*).

## DISCUSSION

*Jansen-Jacobs & Westra (1983)* described *Asterophorum mennegae* Jansen-Jacobs & Westra based on *Oldenburger et al. 415* from Suriname. Later, *Kubitzki (1995)* expanded the circumscription of *Christiana* to include the genera *Asterophorum* (2 spp.) and *Tahitia* (1 sp.). *Kubitzki (1995)* recorded the occurrence of *A. mennegae* in Brazil for the first time. Although Kubitzki mentioned some vouchers from Amazonas and Pará states, the specimens have not been properly identified in herbarium collections and were overlooked by *Secco (2000)*, who reported the occurrence of *Asterophorum mennegae* as the first record of the species for the Brazilian Amazon; nevertheless, this publication was overlooked by subsequent authors. *Kubitzki*'s (*1995*) work is not easily accessible in Brazil and has not yet been made available in online libraries, and this could explain why *Secco (2000)* did not use *Kubitzki*'s (*1995*) circumscription. *Secco (2000)* is not available in online repositories either. *Coutinho (2020)* used online databases such as SpeciesLink and JABOT (http://rb.jbrj.gov.br/v2/consulta.php) and recorded only *C. africana* and *C. macrodon* in the Brazilian Flora 2020 (*Flora do Brasil, 2020*). In addition, the difficulty accessing *Kubitzki (1995)* and *Secco (2000)* may have contributed to the deficient knowledge of *Christiana* until the present moment. The case of *C. mennegae* is an example of overlooked taxa and also highlights the collecting gaps in the Amazon. Similar floristic projects also reported dozens of new records and also overlooked species in the Amazon (*Barbosa-Silva et al., 2016*; *Sierra, Pereira & Zartman, 2019*).

Stomata on seed surfaces are relatively uncommon (*Paiva, Lemos-Filho & Oliveira, 2006*), although recorded for several unrelated plant families (*Jernstedt & Clark, 1979*), and the presence of such structures in *C. mennegae* is the first report for the Brownlowioideae. On the other hand, the presence of stomata in Malvaceae seeds has been reported in other subfamilies, such as Sterculioideae, Bombacoideae and Malvoideae (*Corner, 1976*). The morphological variation on seed surface and indumentum in other groups of Brownlowioideae has already shown to be of taxonomic importance, such as presence of trichomes on the seeds in *Jarandersonia* (*Chung, Tan & Soepadmo, 2012*). Besides being a stable character, the presence of trichomes in seeds may contribute to the taxonomy of the group, and was used in conjunction with other seed surface characters to recognize a new taxon in *Jarandersonia* from Borneo (*Tan, Chung & Soepadmo, 2011*). However, there is an example in the genus *Iris* L. (Iridaceae) in which the occurrence of seed stomata was not a useful taxonomic character (*Wang & Hasenstein, 2016*).

Five trichome types were found in *Christiana mennegae* and among them, the glandular trichomes on the abaxial surface of the leaf-blades, were also seen in species of *Luehea* Willd.,

*Lueheopsis* Burret, *Vasivaea* Baill., *Triumfetta* L. and *Mollia* Mart. (Malvaceae-Grewioideae), as well as in *Heliocarpus popayenensis* H.B.K. (Malvaceae-Grewioideae) and *Christiana africana* (*Westra, 1967*), where the indument beneath the leaves is markedly different in comparison with that of the petioles and branches, as related here for *C. mennegae*. *Westra (1967)* performed a broad micromorphological study in *Apeiba* and provided valuable data about distribution and trichome types for that genus. *Tan, Chung & Soepadmo (2011)* and *Chung, Tan & Soepadmo (2012)* used the trichome type on the leaf, fruit and seed, associated to other characters to distinguish between species of *Jarandersonia* (Malvaceae-Brownlowioideae). *Nurhanim, Noraini & Chung (2010)* studied the trichomes diversity in 18 species of *Pentace* Hassk. (Malvaceae-Brownlowioideae) and demonstrated the systematic taxonomic value of 15 trichome types found.

Ultimately, it was not possible to confirm the monophyly of *Christiana* due to the low sampling of taxa and conflicting statistical support obtained (BS = 86; PP = 0.78). On the other hand, our data supported the inclusion of *Christiana*, *Carpodiptera* and *Berrya*, which present fused calyx-lobes (*Whitehouse et al., 2001*), in a strongly supported clade, as previously indicated by *Alverson et al. (1999)*, *Bayer et al. (1999)*, *Richardson et al. (2015)* and *Hernández-Gutiérrez & Magallón (2019)*. Although a close relationship between *Christiana* and *Carpodiptera* is reasonable, as they are the only Brownlowioideae genera with unisexual flowers (*Bayer & Kubitzki, 2003*), increasing both the sampled species and the number of DNA regions is crucial to clarify the relationships within this clade.

The availability of morphological and molecular data helps to accelerate the delimitation and identification of species (*Ramalho et al., 2018*), thus allowing to rapidly quantify plant diversity in the Amazon. Other Amazonian species may prove to have similar wide range expansions when more collecting effort is employed in remote areas, more investment in training botanists (*Ahrends et al., 2011*), more herbarium collections are digitized and published online, and missing literature is eventually made available electronically.

## CONCLUSIONS

The example of *Christiana mennegae* highlights the need for continuous investment in the digitization of biological collections, investment in local capacity building for botanists and the availability of online literature for the study of Amazonian biodiversity. The new occurrences recorded here include samples collected many years ago but that were either overlooked or not identified, and therefore an already accepted species had escaped the radar of the Brazilian Flora until now. Our intention was to improve the knowledge to raise the profile of *C. mennegae* by adding data of its micromorphology and a preliminary phylogenetic perspective. It is possible that the discovery of the stomata in the seed surface in this subfamily may be interesting to contribute towards the elucidation of the evolution of morphology in Malvaceae.

## ACKNOWLEDGEMENTS

The authors thank, Ana Correa from Laboratório de Microanálises do Instituto de Geociências (IG) in UFPA for the help in taking SEM images. To João Silveira, who

prepared the line drawing. To Caroline Oliveira Andrino for helping in phylogenetic analyses and comments on an early version of the paper. To Raquel Negrão for comments regarding conservation status. To the Curators of the collections studied, especially Helena J.R. Souza (IAN), Pedro L. Viana (MG) and Lourival Tyski (HCJS).

### Funding

CNPq provided a project grant to Rafael G Barbosa-Silva (380010/2020-8). Daniela C Zappi and Guilherme Oliveira hold CNPq productivity grants and Thales Silva Coutinho has a Ph.D scholarship (process x' / 2017-0). The funders had no role in study design, data collection and analysis, decision to publish, or preparation of the manuscript.

### Grant Disclosures

The following grant information was disclosed by the authors:
CNPq.

### Competing Interests

The authors declare there are no competing interests.

### Author Contributions

- Rafael G. Barbosa-Silva conceived and designed the experiments, performed the experiments, analyzed the data, prepared figures and/or tables, authored or reviewed drafts of the paper, and approved the final draft.
- Thales Silva Coutinho, Santelmo Vasconcelos, Daniela C. Zappi conceived and designed the experiments, analyzed the data, authored or reviewed drafts of the paper, and approved the final draft.
- Delmo Fonseca da Silva and Guilherme Oliveira conceived and designed the experiments, authored or reviewed drafts of the paper, and approved the final draft.

### Field Study Permissions

The following information was supplied relating to field study approvals (i.e., approving body and any reference numbers):

Floresta Nacional de Carajás approved this study [63324-1 (Sisbio)].

### DNA Deposition

The following information was supplied regarding the deposition of DNA sequences:

The sequences are available in the Supplementary Files and at GenBank: MT741784 to MT742086.

### Data Availability

The data (maximum likelihood tree) are available in the Supplemental Files.

## Supplemental Information

Supplemental information for this article can be found online at http://dx.doi.org/10.7717/peerj.12244#supplemental-information.

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
