# Peer review of "Preliminary placement and new records of an overlooked Amazonian tree, Christiana mennegae (Malvaceae)"

_PeerJ, doi:10.7717/peerj.12244_

## Round 0.1 · original submission · Major Revisions

Thank you for the manuscript. both reviewers offer useful and constructive suggestions on how to modify the paper. I look forward to seeing a revised version.

Reviewer 1 ·

Basic reporting

see below

Experimental design

see below

Validity of the findings

see below

Additional comments

I believe the manuscript would benefit from a narrower focus. The manuscript seems to be making several different arguments and I would be tempted to jettison the first of the two main arguments 1) The authors argue that in the various recent compilations of data on Brazilian plants, some taxa have been overlooked. (This is, in some respects, trivial). 2) Among the overlooked taxa is Christiana mennegae and not only do the authors add it to the Brazilian flora (again), they also sequence it and place it in a phylogenetic context, and they look at the trichome morphology of the same species. In addition, they extend its range to Amazonian Peru.

The argument about collecting gaps and taxa being overlooked is based on one example. If the authors are to do justice to this argument, they need to marshal additional data otherwise their argument comes across as anecdotal.

Title. The “identity” of Christiana mennegae is not something that is or was in question and the word “identity” should be suppressed. Usually when a taxonomic paper mentions “identity”, the plant in question is or was grossly misinterpreted (e.g., placed in the wrong family).

The “new records” for Brazil, if they consist only of Amazonas and Pará collections, are not new. These state records were reported already. What is new, is the extension of the range of the species to Amazonian Peru.

Pantropical – I think of Pantropical taxa as occurring in the Americas, Africa, and Asia. A species that is found in the American and African tropics only is not pantropical.

Distinguishing the American species of Chirstiana – The text is confused. This information could be expressed well as either a revised key (following Kubitzki, 1995) or a table.

Description of trichomes. I am not sure that multi-angulate captures what you are trying to convey. When I look at your images, I see some stellate hairs with the arms (comparatively few in number) in a single plane (rotate) and others with many more arms in several planes. I would describe these latter as tufted, rather than angulate.

References. These need to be checked carefully. There are many spelling errors and incomplete citations. Also, the punctuation is not standardized. Some titles are consistently lowercase while others are mixed case, etc.

Illustrations. The illustrations are all well done and informative.

Annotated reviews are not available for download in order to protect the identity of reviewers who chose to remain anonymous.

Reviewer 2 ·

Basic reporting

No comment.

Experimental design

No comment.

Validity of the findings

No comment.

Additional comments

The authors of this study explore the taxonomy, micromorphology, and phylogenetic relationships of poorly known species, Christiana mennegae, in an almost equally poorly known genus of Malvaceae. This is a short paper. It includes new observations and so certainly makes a contribution to our knowledge of this species. However, the paper seems a little insubstantial and important information is missing from the methods and results. The editor needs to make the decision whether this study is significant enough to warrant publication. The paper is very clearly written, although there are several minor problems with the English (grammar and idiomatic expressions). I strongly encourage the authors to find a good English-speaking editor to help. More detailed comments are below.

Intro:

The authors begin by making a compelling argument that knowledge of the flora of the Amazon region is incomplete and that additional studies, and new kinds of studies, are needed. This is nice writing, but this subject is not revisited in the rest of the paper. How does more information on one species help our understanding of the Amazon region? How does this study illustrate and fix the general problem? What strategies would prevent other species from being similarly overlooked in the Flora of Brazil? Do they predict other species will have similar enormous range expansions if/when there is more coordination about taxonomists?

The paper would benefit from a clear summary of the genus Christiana in the introduction. How many species are in the genus, their distribution, generic limits, any question or investigation into its monophyly. All of the information is scattered in the manuscript, but it needs to be presented clearly in the introduction. Also, if Christiana africana has the same distribution as Ceiba pentandra (tropical America and tropical Africa), it should not be called pantropical. The authors may want to track down Kubitzki’s treatment of Malvaceae in the 2003 Families and Genera of Vascular Plants.

For the taxonomic treatment, how many specimens were seen (either physical specimens or digital images)? How many new collections were made? Fig. 1 shows four dots. What does each dot represent? A collection? A population? Was the map prepared only using herbarium specimens, or are new collections included? Can the authors make estimates of population sizes based on their field work? If the species is dioceious, how many staminate specimens were observed, and how many carpellate?

How was conservation status determined?

SEM: There is no information on how SEM was performed. Were structures reconstituted from dry specimens? And if so, how? How were the mounts prepared? Did they go through a dehydration series? Were they sputter-coated? Was the SEM under full vacuum or low vacuum? What structures were examined with SEM, and why? In the methods, the authors state “vegetative and reproductive structures,” which includes pretty much the entire plant. But in the results, they only discuss trichomes and stomata. Was there an a priori reason to examine trichomes and stomata, or were these the only observable traits? The micrographs in Fig. 3 are beautiful, but there is so little known of the micromorphology of Brownlowioideae that is is difficult to interpret the significance of these results. How many specimens
were examined, and how many structures (leaves, flowers, fruit) per specimen? Were leaves mature or immature? All of this information is important to evaluate how robust the observations are.

Phylogenetic analysis: The authors add two short sequences (rbcL and ITS2) from C. mennegae to a previously published dataset of Malvaceae. There are several issues here. It would be good to see a hypothesis or rationale for performing a phylogenetic analysis. Is there a reason to think that Christiana is not monophyletic? That C. mennegae is not a member of Brownlowioideae? The dataset that the authors added their sequences to is enormous, and far larger than they need (unless there’s a possibility that C. mennegae is not in Brownlowioideae). Hernandez-Gutierrez and Magallon reanalyzed data that had been previously published by other authors, so it is not accurate to say that they published sequences from 239 species of Malvaceae. I don’t know the details of this matrix, but I’m sure from the description that there is a large amount of missing data. None of that is reported here. My suggestions: State your hypothesis for the phylogenetic analysis. Construct a matrix that is a reasonable test of your hypothesis. This probably means Brownloidoideae and a few outgroups from other subfamilies of Malvaceae. Exclude gene regions for which there are large amounts of missing data. Give more detail on the final dataset, your analyses, and the results (seven lines for the results is a red flag). I appreciate the authors’ caution in interpreting their phylogenetic results. But, given that C. mennegae and C. africana appear to only have the slow-evolving rbcL sequences in common, it is quite remarkable that their trees show these two species as sister taxa. Their monophyly is not strongly supported, but I would not expect it to be given the data.

Figures:

The Fig. 1 legend needs more detail. How many collections is this map based on? Do these represent individual collections, known populations? We need some numbers here. The compass and scale bars would be more readable if they were also in white.

Fig. 2 is lovely, but the legend should include information of the specimen that the illustration is based on.

Fig. 3 is also lovely, but also needs specimen information. I don’t like the colorization, and I think it makes it difficult to see detail, especially of the stomata and glandular trichomes. It helps a little, perhaps, in Fig. 3a, but not at all in b-h.

Fig 4: See my comments on the phylogenetic analysis.

Fig. 5 Again, lovely, but are the colors enhanced in c-e? If so, it should be stated. Information on this particular specimen should be given. I don’t think you need both Fig. 5d and 5e, and the cropping of 5e is problematic.

---

## Round 0.2 · Major Revisions

There remain very considerable concerns about this manuscript. It is a very lengthy paper adding only limited new detail on an admittedly poorly known species. One reviewer has now recommended rejection and I would concur with this in the manuscript's current form. However, given the under reporting of many tropical taxa, and the lack of knowledge of this species for conservation assessment I would like to offer one more chance at major revision. I would suggest you shorten the manuscript substantially and reduce the phylogenetic study to species immediately related to this genus. You should include the reduced DNA sequence alignment as supplementary material and report more fully the support for monophyly of the genus. You should also add an explicit review of the conservation status of the species based on the new distribution data. The currently published IUCN red list entry is quite out of date and this addition might tip the balance between rejection and acceptance of the manuscript in the eyes of reviewers.

Reviewer 1 ·

Basic reporting

See below

Experimental design

See below

Validity of the findings

See below

Additional comments

I read the revised manuscript and the authors’ response to points the two reviewers raised in their reviews. While the authors addressed the easily resolvable problems, they avoided the more serious ones. The current title promises “phylogenetic placement” of Christiana mennegae while the abstract and introduction suggest that the study addresses large scale biodiversity and digitization issues. The manuscript does not deliver on these promises. A single molecular sequence for an infrequently-collected species as well as some original morphological (SEM) observations are presented. While there is value in these observations, they are “preliminary.”

---

## Round 0.3 · Minor Revisions

It has been challenging to get timely reviews for your manuscript which has resulted in me changing reviewers from version to version. However, you have made many of the modifications needed and I think the addition of input from these reviews should bring your paper to a point where it can go through a final round of edits. Thank you for the work you have put in so far.

Reviewer 3 ·

Basic reporting

I only have suggestions in order to improve the content and correct minor errors.

Some suggestions are made below:
- Line 37: Include “Brownloioideae” in keywords.
- Line 62: The citation for De Candolle (1824) must be included in the References.
- Line 74: “Secco (2000) recorded Asterophorum mennegae…”. Include after “Asterophorum mennegae” the term “(= C. mennegae)”?
- Line 112-113: “It was not necessary to critical-point dry them.” I think that this sentence is not necessary because the authors previously mentioned on line 109 that the samples were dried.
- Line 186-187: The mentioned Table includes results from this manuscript, but also results compiled from the literature. The results obtained from the literature must be indicated and referenced.
- Line 207: This information must be transferred to M&M, not mentioned here.
- Line 227-231: The description of trichomes is in Supplementary Material, but it’s important to note the types of trichomes in this paragraph.
- Line 228: It’s not necessary repeat the citation of “Theobald et al. 1979.” This information is already noted in M&M.
- Line 245, end of Results: Figure 5 wasn't explored. In the absence of information about this figure, I suggest transferring it to Supplementary Material.
- Line 263: The two cited authors must be included in the REFERENCES.
- Line 276-277: Glandular trichomes were also seen in the petioles. See Figure 3A. The description of glandular trichomes must also be included in the Supplementary Material.
- Legend, Figure 2: In C, it's difficult to see the mentioned trichomes. I suggest an insert like Figure F.
- Legend, Figure 3: Figure B mentioned the presence of stomata, but this structure wasn’t mentioned in the Results.
- Legend, Figure 3: In E, the magnification of the image makes it impossible to see the type of trichome.
- Legend, Figure 3: In F, it would be appropriate to provide an image with the same magnification as that in Figure 3D. In the presented figure, it is impossible to view the trichome type.
- Legend, Figure 3: Arrows cannot be seen in the figure.
- In the Electronic Supplementary Material:
o Material and Methods: In the first paragraph, line 4, include in the References the citation of Ramalho et al. 2018.
o Results: In Table 1, include the source of the information not original to this manuscript. The data about the geographic distribution of all cited species must also be included.
o Micromorphology study: first paragraph –
 Line 1-3: The glandular trichomes weren’t mentioned or even described.
 Line 4-5: The stellate-rotate trichomes aren’t visible in Figure 3B. “fig 3C” should be deleted.
 Line 8-9: Figures 3A-C are about vegetative structures (leaf and petiole), not reproductive. The reproductive structure is represented in Figures 3D-F.
 Second paragraph: How is this text related to “Micromorphology study”?

Experimental design

no comment

Validity of the findings

no comment

Additional comments

I would like to congratulate the authors for their research. The core of the article is creative and obtained from data that could not even be considered.

The images presented are of high quality, and the analyses were performed equally well.

·

Basic reporting

The English writting seems to be sufficient, while the writting can be improved in some parts.

The article does not include sufficient background on broader field of knowledege. Relevant prior literature was not critically analyzed and referenced especially on morphology of Malvaceae that should aid in data interpretation and discussion of results. I consider the manuscript needs to be better conceptualized and discussed from a morphological view. I have made suggestions in the attached files including some references.

The manuscript is nicely illustrated. Figures are relevant to the content, but should be better described and labeled (some suggestions included in attached files). Nevertheless, the map should include all species occuring in that geographic region to allow comparison (as done in the tree, table, text, etc.; it is comparative biology).

Unfortunately I could not check all raw data (I do not remember seeing a data file used for GEOCAT analysis).

Experimental design

The manuscript includes original primary research. Research question is relevant and the knowledge gap was identified and the statements were made as to how the study contributes to filling that gap, but I consider the study could be better designed.

There seems to be some bias to digitalization of both herbarium specimens and literature (I think it should be one aspect to be discussed but not the main focus of the mss and the introduction). Authors focused on the absence of the focal species on generic floristic lists and recent monographs linking to digitization of both herbarium specimens and literature but they not discuss it more broadly to include the importance of specialists in Malvaceae from Brazil and taxonomy in general. It would be expected that specialists, rather than general floristic papers like ter Steege et al. or Cardoso et al., could detect this plant taxon in herbaria. Consequently, the taxon would appear in more general floristic lists.

Thus I think it would be more meaningful and helpful to discuss data in the light of the need of investment to training botanists (e.g., Ahrends et al., 2011 in Biological Conservation) and examine extant herbarium collections (e.g., Bebber et al. 2010 in PNAS) to disclose diversity of plants and avoid overlooking species.

The obtained sequences for Christiana mennegae are welcome to the phylogeny of Malvaceae but they would be unnecessary to place C. mennegae as related to C. africana (and authors should NOT state they are sister groups because sampling of the study does not allow to test such a relationship as only these two species were sampled; additional comments on attached files). This might be achieved with morphology only since both species are clearly related based on flower and fruit morphology. Moreover, the molecular focus and relative hard work on lab and analyses seem to have biased authors’ view at the expense of morphological aspects of Christiana mennegae in the context of Malvaceae. The phylogenetic framework in the mss was not used to discuss the morphology, evolution, or any other aspects.

Therefore, rigorous investigation seems to have been performed to a high technical standard regarding molecular part of the study but not the morphological part nor the conservation status assessment (two of the my main concerns). Literature on Malvaceae morphology as well as IUCN (2012) criteria should be investigated more deeply and thoroughly as well as more critically by this study. For example, I could not find justifications for the authors having considered the very similar fruits of C. africana and C. mennegae, respectively, apocarpous and syncarpous.

Also, I could not find justifications for the authors 1) having used EOO (extension of occurrence) but not AOO (Area of occupancy) which is also calculated by the tool used (GEOCAT) by them, and 2) having overlooked the need for selecting at least two of the three required conditions for IUCN criteria they used for conservation assessment (i.e., “B. Geographic range”) that can include number of locations of occurrence for which they presented data. Please see additional comments on attached files.

Validity of the findings

Some conclusions regarding, as for instance, the monophyly of Christiana, the occurrence of stomata in seeds surface and the inexistence of glandular trichomes on other four species of Christiana, are not connected to the original question investigated.

Some conclusions are fragile, like, for instance, about inexistence of sympatry between Christiana africana and C. mennegae because no (comprehensive) data on geographic distribution (from herbarium records and literature) of Christiana taxa was compiled, presented or analysed. For example, Tschá et al. (2002) cited herbarium specimens of C. africana from municipality of Rio Vermelho in the Pará state that is c. 50 km east from collections of C. mennegae in the municipality of Canaã dos Carajás.

Additional comments

I consider the manuscript is poorly conceptualized from a morphological view. Important literature on Malvaceae morphology was overlooked and not referenced in the introduction and discussion. This could contribute to a more detailed morphological description and a deeper discussion of their data. I would expect to find a critical review of the literature about the placement of Christiana in Malvaceae and Brownlowioideae. It was not clear from introduction what morphological evidence lead authors to identify material as the belonging to subfamily Brownlowioideae of the Malvaceae (e.g., fused sepals? Presence of petals? Free stamens? Staminal thecae divergent at the base? Presence of staminodes?). It would be informative and helpful to have (diagnostic) morphological characterization of this poorly known subfamily in Brazil.

Also, I have found several inconsistencies between main text and table in the manuscript regarding important morphologic aspects (including diagnostic characters) as well as main text and figure legends (e.g., classification/nomenclature of trichome types). Please see additional comments on attached files.

The IUCN conservation status asssessment was not correct. The Least Concern category applies to widespread and abundant taxa that does not qualify for Criticallly Endangered, Endangered, Vulnerable or Near Threantened. Altough authors wrote the focal taxon “spans throughout Amazon basin”, their map and specimens examined do show basically four locations sensu IUCN (2012: p. 13) based on seven collections. I think C. mennegae would qualify for VU B2ab(ii,iii).

Also, I do consider that (second-step) typification of Asterophorum is necessary to meet requirements of ICN (Turland et al., 2018). But I did not get access to the protologue and related publications. Please see comments in attached files. Authors argue strongly in favor of making available digital both literature and herbarium specimens because. I could not get literature authors stated are difficult to obtain and so I have appreciated if they had made Kubitzki (1995) and Secco (2000) available to referees to facilitate reviewing the mss.

---

## Round 0.4 · accepted · Accept

There are a few corrections remaining to be made but otherwise the paper is ready for acceptance.
Line 105 - correct alsooverlooked to also overlooked
Line 238 - correct gamossepalous to gamosepalous
Thank you for taking the time to incorporate the corrections suggested by the reviewers.

The Section Editor also recommended the following edits:

EDITS
LINE NO: / BEFORE / AFTER / [COMMENTS]
LINE 27: / micromorphological / micro-morphological / [since macro- preceeds.]
LINE 64: / stamens numerous, free, anthers / . / [punctuation was a bit strange; please re-evaluate.]
LINE 77: / alsooverlooked / also overlooked / [.]

Reviewer 3 ·

Basic reporting

no comment

Experimental design

no comment

Validity of the findings

no comment

Additional comments

no comment